# Reparameterization Flow Policy Optimization

**Hai Zhong**[1]   **Zhuoran Li**[1]   **Xun Wang**[1]   **Longbo Huang**[1]

## Abstract

Reparameterization Policy Gradient (RPG) has emerged as a powerful paradigm for model-based reinforcement learning, enabling high sample efficiency by backpropagating gradients through differentiable dynamics. However, prior RPG approaches have been predominantly restricted to Gaussian policies, limiting their performance and failing to leverage recent advances in generative models. In this work, we identify that flow policies, which generate actions via differentiable ODE integration, naturally align with the RPG framework, a connection not established in prior work. However, naively exploiting this synergy proves ineffective, often suffering from training instability and a lack of exploration. We propose Reparameterization Flow Policy Optimization (RFO). RFO computes policy gradients by backpropagating jointly through the flow generation process and system dynamics, unlocking high sample efficiency without requiring intractable log-likelihood calculations. RFO includes two tailored regularization terms for stability and exploration. We also propose a variant of RFO with action chunking. Extensive experiments on diverse locomotion and manipulation tasks, involving both rigid and soft bodies with state or visual inputs, demonstrate the effectiveness of RFO. Notably, on a challenging locomotion task controlling a soft-body quadruped, RFO achieves almost $2\times$ the reward of the state-of-the-art baseline.

## 1. Introduction

Reparameterization Policy Gradient (RPG) parameterizes the policy as a differentiable mapping from a source noise distribution to actions (Kingma & Welling, 2014; Rezende et al., 2014), allowing gradients to be backpropagated directly through the trajectory. Compared to likelihood-ratio estimators such as REINFORCE (Williams, 1992), RPG generally exhibits lower variance and thus higher sample efficiency (Gao et al., 2024; Xing et al., 2025). These advantages have fueled a surge of robotic applications built on differentiable simulators, spanning quadrupeds to quadrotors (Luo et al., 2025; Zhang et al., 2025b; Heeg et al., 2025; Schwarke et al., 2025; Pan et al., 2025).

Yet RPG has so far been studied mostly in the context of Gaussian policies, which restricts the family of action distributions the policy can express. In contrast, flow policies trained via imitation learning have recently demonstrated remarkable success in robotics (Black et al., 2025a; 2026; 2025b; Lipman et al., 2023), owing to their superior expressiveness, implementation simplicity, and fast inference. To overcome the limitations of demonstrations, there has been a surge of research interest in leveraging reinforcement learning (RL) to enable flow policies to learn directly from their own experience (Intelligence et al., 2025; McAllister et al., 2025; Zhang et al., 2025c;a; 2026; Lv et al., 2025).

In this work, we identify the key insight that flow policies inherently parameterize a differentiable mapping from source noise to actions, thus naturally aligning with the RPG framework. Leveraging this synergy, we view the flow generation process as an integral part of the rollout trajectory, enabling policy gradients to be computed by backpropagating directly through both the ODE integration and system dynamics, a novel connection not established in prior work. However, directly optimizing flow policies with RPG fails to yield strong performance, primarily due to optimization instability and insufficient exploration.

Hence, we propose **Reparameterization Flow Policy Optimization (RFO)**, an on-policy RL algorithm optimizing flow policies with RPG. Beyond the core RPG mechanism, we introduce two regularization terms crucial for stable training. The first leverages actions sampled during rollouts as targets for Conditional Flow Matching (CFM) to stabilize policy updates, while the second utilizes targets from the uniform distribution to explicitly encourage exploration. Additionally, we propose a variant of RFO incorporating an action chunking mechanism (Black et al., 2025a;b; Li et al., 2025b). We evaluate our method across a diverse suite

---

[1]Institute for Interdisciplinary Information Sciences (IIIS), Tsinghua University. Correspondence to: Longbo Huang <longbohuang@tsinghua.edu.cn>.

*Proceedings of the 43rd International Conference on Machine Learning*, Seoul, South Korea. PMLR 306, 2026. Copyright 2026 by the author(s).

of locomotion and manipulation tasks, involving both rigid and soft bodies, using state or visual inputs. Notably, RFO achieves nearly $2\times$ the reward of the state-of-the-art (SOTA) baseline on a challenging soft-body quadruped locomotion task with a high-dimensional action space ($\mathbb{R}^{222}$).

RFO represents a meaningful contribution to both the flow-based RL and RPG communities. From the perspective of flow-based RL, RFO uniquely combines the strengths of distinct paradigms: it inherits the high sample efficiency of RPG while eliminating the need for intractable action log-likelihood computations. Simultaneously, as an on-policy approach, RFO seamlessly integrates with massively parallel differentiable physics simulators, such as Rewarped and DFlex (Xing et al., 2025; Xu et al., 2021; Georgiev et al., 2024; NVIDIA et al., 2025), to significantly accelerate training.

To summarize, our contributions are threefold: (i) We propose a novel paradigm for training flow policies using Reparameterization Policy Gradients, unlocking high sample efficiency without approximating intractable likelihoods. (ii) We introduce Reparameterization Flow Policy Optimization (RFO), an algorithm equipped with novel regularization mechanisms for stability and exploration, further extended with an action-chunking formulation. (iii) We conduct extensive experiments on differentiable simulators, demonstrating that RFO consistently achieves strong performance across challenging rigid and soft-body tasks compared to existing baselines.

## 2. Related Work

### 2.1. Online Reinforcement Learning Algorithms for Flow and Diffusion Policy

Online RL algorithms for flow and diffusion policies can be categorized as off-policy or on-policy. Off-policy algorithms rely on Q-functions to train generative policies (Psenka et al., 2024; Wang et al., 2025a; Celik et al., 2025; Lv et al., 2025; Ma et al., 2025; Chen et al., 2025; Dong et al., 2025; Yang et al., 2023; Zhang et al., 2025c; Ding et al., 2024; Wang et al., 2025b; Park et al., 2025), offering the advantage of bypassing the calculation of action log-likelihoods. One approach for policy optimization involves backpropagating Q-gradients to the noise prediction or vector field network (Wang et al., 2024; 2025a; Lv et al., 2025; Wang et al., 2025b; Zhang et al., 2025c; Celik et al., 2025). This is achieved by reparameterizing the flow or diffusion policy as a transformation from sampled noise to actions, which allows for the direct backpropagation of Q-function gradients through this differentiable transformation to the policy network. Another class of approaches exploits the intrinsic connection between the score function (or velocity field) and Q-functions (Psenka et al., 2024; Dong et al., 2025; Akhound-

Sadegh et al., 2024; Ding et al., 2024). For instance, (Psenka et al., 2024) proposes matching the score functions to Q-function gradients. Meanwhile, (Dong et al., 2025; Ma et al., 2025; Ding et al., 2024) utilize a Q-weighted regression loss to train the noise prediction network, whereas (Chen et al., 2025) trains the velocity network via a Q-weighted conditional flow matching loss.

On-policy RL algorithms for flow and diffusion policies (Ding et al., 2025; Ren et al., 2025; Zhang et al., 2025a; Yang et al., 2025; McAllister et al., 2025) must address the challenge of approximating otherwise intractable action log-likelihoods. One prevalent approach is to treat the sampled noise as an augmented state, which enables the calculation of action likelihoods conditioned on the noise. For instance, DPPO (Ren et al., 2025) constructs a two-level Markov Decision Process that propagates policy gradient signals to each denoising step. Similarly, NCDPO and Reinflow (Yang et al., 2025; Zhang et al., 2025a) treat the action generation as a deterministic process by conditioning on all sampled noise, and additionally train a noise injection network to compute action log-likelihoods. Alternatively, FPO (McAllister et al., 2025) approximates action log-likelihoods via the conditional flow matching loss. GenPO (Ding et al., 2025) introduces a specialized doubled dummy action generation mechanism, which enables the calculation of exact action log-likelihoods. In contrast, RFO entirely bypasses likelihood computation by leveraging reparameterization gradients through differentiable dynamics, which typically yields lower-variance gradient estimates than REINFORCE-like policy gradients (Mohamed et al., 2020).

### 2.2. Reparameterization Policy Gradient

Reparameterization Policy Gradient exploits the reparameterization trick (Kingma & Welling, 2014; Rezende et al., 2014; Mohamed et al., 2020) to render the action sampling process differentiable. By defining the action as a deterministic transformation of a noise variable drawn from a fixed source distribution, RPG enables the computation of low-variance policy gradients by backpropagating directly through the system dynamics. These dynamics are typically provided by either differentiable simulators (Xu et al., 2021; Hu et al., 2020; Xing et al., 2025) or learned world models (Hafner et al., 2020; Amos et al., 2021; Li et al., 2025a). Consequently, RPG generally yields gradient estimates with significantly lower variance compared to likelihood-ratio estimators like REINFORCE (Mohamed et al., 2020; Sutton et al., 1999). Prominent RPG algorithms, such as SHAC and AHAC (Xu et al., 2021; Georgiev et al., 2024), further stabilize training by truncating the backpropagation horizon and leveraging value function gradients to estimate long-term returns. While prior work has investigated the use of multimodal policies within RPG (Huang et al., 2023), the integration of modern expressive generative models,

specifically diffusion and flow matching, remains largely unexplored. RFO bridges this gap by introducing the first RPG-based framework tailored for flow policies.

## 3. Preliminaries

### 3.1. Reinforcement Learning Problem Formulation

We formulate the reinforcement learning (RL) problem as a Markov Decision Process (MDP) (Sutton & Barto, 2018), defined by the tuple $(\mathcal{S}, \mathcal{A}, p, r, \rho_0, \gamma)$. Here, $\mathcal{S}$ denotes the state space, and $\mathcal{A}$ denotes the action space. The transition function $p(s'|s, a)$ defines the probability density of transitioning to the next state $s' \in \mathcal{S}$ given the current state $s \in \mathcal{S}$ and action $a \in \mathcal{A}$. The term $r(s, a)$ represents the reward function, $\rho_0$ denotes the initial state distribution, and $\gamma \in [0, 1)$ is the discount factor. In this work, we consider a continuous and **bounded** action space $\mathcal{A}$. The objective of RL is to maximize the expected discounted cumulative reward:

$$J(\theta) = \mathbb{E}_{\tau \sim \pi_\theta} \left[ \sum_{t=0}^{\infty} \gamma^t r(s_t, a_t) \right], \tag{1}$$

where $\tau = (s_0, a_0, s_1, a_1, \dots)$ denotes a trajectory sampled from the distribution induced by the policy $\pi_\theta$, with $s_0 \sim \rho_0$.

### 3.2. Flow Model and Flow Policy

**Flow Models and Flow Matching.** Flow models (Lipman et al., 2023; Chen et al., 2018; Albergo & Vanden-Eijnden, 2023; Liu et al., 2023) constitute a class of generative models that transform samples from a source distribution $x_0 \sim p_0$ (e.g., standard Gaussian) to a target distribution $x_1 \sim p_1$ via an invertible flow map $\phi : \mathbb{R}^d \times [0, 1] \to \mathbb{R}^d$. This flow map is induced by a time-dependent, neural-network-parameterized vector field $v_\theta(x, u)$ (Chen et al., 2018), defined by the ordinary differential equation (ODE):

$$\frac{d}{du} \phi(x_0, u) = v_\theta(\phi(x_0, u), u), \tag{2}$$

where $\phi(x_0, 0) = x_0$, $u$ denotes the flow time, and $p_0(x_0)$ denotes the density of the source distribution. To learn the parameterized vector field $v_\theta$, the Conditional Flow Matching (CFM) objective is proposed (Lipman et al., 2023). Specifically, CFM constructs a linear interpolation path conditioned on the source noise $x_0$ and the target sample $x_1$, where the intermediate state is given by:

$$x_u = (1 - u)x_0 + ux_1, \quad u \in [0, 1]. \tag{3}$$

Consequently, the CFM objective is defined as:

$$\mathcal{L}_{\text{CFM}}(\theta)$$
$$= \mathbb{E}_{u \sim \mathcal{U}[0,1], x_0 \sim p_0, x_1 \sim p_1} \left[ \| v_\theta(x_u, u) - (x_1 - x_0) \|^2 \right], \tag{4}$$

where $\mathcal{U}[0, 1]$ is the uniform distribution.

**Flow Policy.** Flow models serve as a powerful policy class for generating actions (Black et al., 2025a; Intelligence et al., 2025; Black et al., 2025b) from potentially complex, multimodal distributions. To define a state-conditioned policy, the vector field $v_\theta(x, u|s)$ is conditioned on the current state $s \in \mathcal{S}$. An action $a$ is generated by first sampling $x_0$ from the source distribution $p_0$, and subsequently solving the following ODE from $u = 0$ to $u = 1$:

$$\frac{dx}{du} = v_\theta(x, u|s), \quad \text{with } x(0) = x_0. \tag{5}$$

The core challenge in utilizing flow policies for RL lies in propagating reward signals to optimize the vector field $v_\theta$.

### 3.3. Reparameterization Policy Gradient

RPG is grounded in the reparameterization trick (Kingma & Welling, 2014; Rezende et al., 2014). The key idea is to learn a deterministic transformation that maps a fixed source distribution into the desired action distribution. Concretely, with noise $\epsilon$ drawn from the source distribution, the action is generated by a learned function $a = f_\theta(\epsilon; s)$, typically a neural network, conditioned on the state $s$. To date, the literature has mostly instantiated this transformation as a Gaussian policy (Mohamed et al., 2020; Xu et al., 2021; Georgiev et al., 2024; Xing et al., 2025; Gao et al., 2024), mapping a standard Gaussian source to a state-dependent Gaussian distribution.

Following previous RPG work (Xu et al., 2021; Georgiev et al., 2024; Xing et al., 2025), we focus on deterministic environment dynamics $s_{t+1} = g(s_t, a_t)$ and assume environment dynamics and the reward functions are differentiable.

Leveraging the reparameterization transformation, RPG computes the policy gradient by backpropagating through the environment dynamics, utilizing the dynamics Jacobians $\frac{\partial s_{t+1}}{\partial a_t}$ and $\frac{\partial s_{t+1}}{\partial s_t}$. The objective function and its gradient are defined as:

$$J(\theta) = \mathbb{E}_{s_0, \epsilon_0, \epsilon_1, \dots} \left[ R(\tau) \right], \tag{6}$$
$$\nabla_\theta J(\theta) = \mathbb{E}_{s_0, \epsilon_0, \epsilon_1, \dots} \left[ \nabla_\theta R(\tau) \right], \tag{7}$$

where $R(\tau) = \sum_{t=0}^{\infty} \gamma^t r(s_t, a_t)$ denotes the cumulative discounted reward for the trajectory $\tau$. Crucially, this expectation is taken with respect to the initial state distribution and the sequence of independent noise variables sampled from the source distribution at each time step.

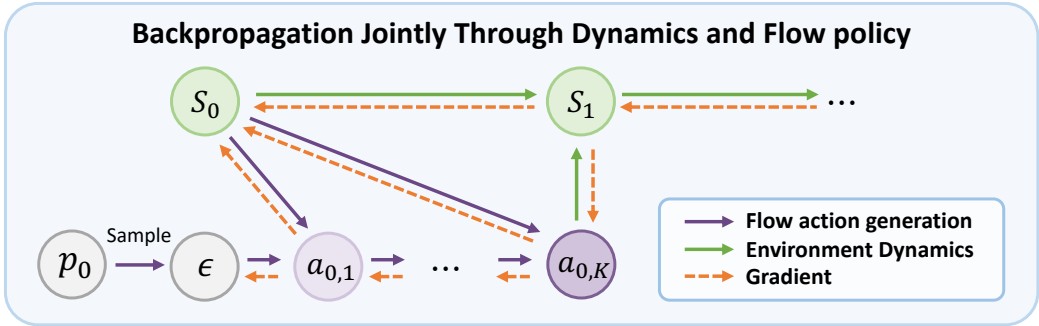

*Figure 1.* RFO optimizes the flow policy by jointly backpropagating through the dynamics and the flow policy.

## 4. Reparameterization Flow Policy Optimization

In this section, we detail the proposed algorithm, Reparameterization Flow Policy Optimization. RFO comprises three key components. First, we derive the optimization of flow policies with RPG. Second, we introduce two regularization terms that are critical for RFO's performance. Finally, we present a variant of RFO that incorporates an action chunking mechanism.

### 4.1. Reparameterization Policy Gradient for Flow Policy

We now detail the first key component of RFO: reparameterization policy gradient for flow policies. We adopt the *discretize-then-optimize* paradigm (Onken & Ruthotto, 2020) for flow policy optimization, consistent with prior studies on reinforcement learning with flow policies (Zhang et al., 2025a; Lv et al., 2025; Zhang et al., 2025c; Ding et al., 2025). Specifically, given the neural-network-parameterized vector field $v_\theta$, we perform numerical integration on the ODE using the Euler method (Eul, 2016). We employ the double-subscript notation $a_{t,k}$, where the subscript $t$ denotes the MDP time step and $k$ indexes the internal discrete integration steps. The generation process is formalized as follows:

$$
\begin{aligned}
a_{t,0} &= \epsilon_t, \quad \text{where } \epsilon_t \sim p_0, \\
a_{t,k+1} &= a_{t,k} + \Delta u \cdot v_\theta(a_{t,k}, u_k|s_t), \quad k = 0, \ldots, K-1, \\
a_t &= a_{t,K},
\end{aligned}
$$
(8)

where $\Delta u = 1/K$ is the step size and $u_k = k \cdot \Delta u$ denotes the $k$-th flow time. Consequently, the Euler integration process explicitly defines a differentiable transformation from the noise $\epsilon_t$ to the action $a_t$. We denote this transformation as $a_t = F_\theta(\epsilon_t; s_t)$. Based on this, we have the following key insight to optimize RFO.

**Key Insight:** Flow policies inherently parameterize a differentiable mapping from source noise to actions. This structure is naturally compatible with the Reparameterization Policy Gradient framework, enabling the computation

of policy gradients via end-to-end backpropagation jointly through the environment dynamics and the flow policy.

An illustration of this insight is shown in Figure 1. To formalize this insight, reparameterization policy gradient for the flow policy is expressed as:

$$
\nabla_\theta J(\theta) = \mathbb{E}_{s_0, \epsilon_0, \epsilon_1, \ldots} \left[ \nabla_\theta \sum_{t=0}^{\infty} \gamma^t r\left(s_t, F_\theta(\epsilon_t; s_t)\right) \right], \quad (9)
$$

where $\epsilon_t$ denotes the noise sampled from the source distribution at time $t$, and $F_\theta$ represents the flow transformation. The gradient is computed via backpropagation through time (BPTT) on the computational graph, which includes both the flow ODE integration and the environment dynamics typically provided by a differentiable simulator or a learned world model.

In practice, backpropagating through long horizons can lead to exploding gradients. Therefore, we follow the Short-Horizon Actor-Critic (SHAC) paradigm (Xu et al., 2021; Georgiev et al., 2024) by truncating trajectories into a batch of short segments to approximately compute gradient estimates for $J(\theta)$. That is, we backpropagate through a truncated horizon:

$$
\nabla_\theta \left( \sum_{t=0}^{H-1} \gamma^t r(s_t, F_\theta(\epsilon_t; s_t)) + \gamma^H V_\omega(s_H) \right), \quad (10)
$$

where $s_0$ represents the start state of the current segment, $V_\omega(s_H)$ is the terminal value estimate to account for future rewards, and $\omega$ denotes the parameters for the value function network. To rigorously estimate $\nabla_\theta J(\theta)$ at each iteration, one should re-sample initial states from $\rho_0$ and restart the rollout. In practice, however, we follow SHAC's procedure and continue from the previous segment's truncated state, which may differ from the initial state distribution, but this works well empirically. Following SHAC, the accumulated gradient is normalized by the product of the batch size and the rollout horizon $H$.

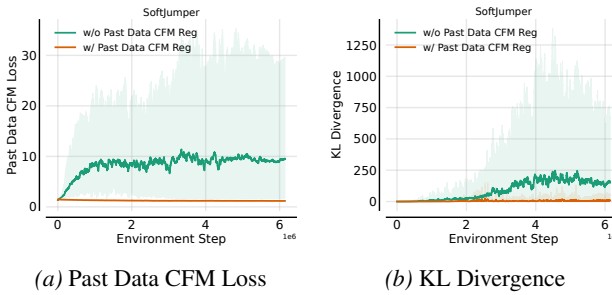

*(a)* Past Data CFM Loss          *(b)* KL Divergence

*Figure 2.* Comparison of RFO training with and without past data CFM regularization on the Soft Jumper task. We monitored the Conditional Flow Matching (CFM) loss of the current policy evaluated on actions sampled from the immediately preceding iteration, as well as the KL divergence between consecutive policy updates.

## 4.2. Regularization Terms

RFO incorporates two regularization terms that are crucial for performance. The first term addresses the stability of policy updates. RPG optimizes the policy by backpropagating gradients through the ODE integration steps, which can potentially disrupt the flow trajectories towards previously generated actions, rendering them unlikely to be re-sampled under the updated policy.

To illustrate this phenomenon, we conducted experiments on the Soft Jumper task using the Rewarped simulator (Xing et al., 2025). We monitored the CFM loss of the current policy evaluated on actions sampled from the immediately preceding iteration. As shown in Figure 2, the CFM loss steadily increases and remains high, indicating that the updated vector field struggles to reconstruct past actions. Furthermore, we approximately tracked the KL divergence between consecutive policy updates (KL divergence approximation details are provided in Appendix E). The large KL divergence clearly signals training instability.

---

**Algorithm 1** Reparameterization Flow Policy Optimization (RFO)

---

1: Initialize $\theta, \omega$ and buffers $\mathcal{D}_{\text{recent}}, \mathcal{D}_{\text{rollout}}$.
2: **for** iteration $i = 1, 2, \ldots$ **do**
3:    Collect short-horizon trajectories with flow policy; update $\mathcal{D}_{\text{recent}}$ and $\mathcal{D}_{\text{rollout}}$.
4:    Compute the RPG gradient for the flow policy via BPTT (Eq. (10)).
5:    Compute regularization CFM loss gradients $\nabla \mathcal{L}_{\text{past}}$ (Eq. (11)) and $\nabla \mathcal{L}_{\text{uni}}$ (Eq. (12)).
6:    Update flow policy with combined and weighted gradients (Eq. (14)).
7:    Update critic $\omega$ for $L$ epochs by minimizing Eq. (13).
8: **end for**

---

Hence, intuitively, we aim to regularize the vector field to

retain paths from the source distribution to these recently sampled actions. We propose the *Past Data CFM Regularization*. This term treats the recent rollouts' actions as target samples and ensures the vector field points towards them. The objective $\mathcal{L}_{\text{past}}$ is defined as:

$$
\begin{aligned}
&\mathcal{L}_{\text{past}}(\theta) \\
&= \mathbb{E}_{\substack{u \sim \mathcal{U}[0,1], \epsilon \sim p_0, \\ (s,a) \sim \mathcal{D}_{\text{recent}}}} \left[ \left\| v_\theta(\psi_u, u|s) - (a - \epsilon) \right\|^2 \right],
\end{aligned} \quad (11)
$$

where $\mathcal{D}_{\text{recent}}$ is a buffer including state-action pairs from the two most recent iterations (including the current iteration's rollout data), and $u$ is the flow time. A discussion of the design of $\mathcal{D}_{\text{recent}}$ is in Appendix F. Here, $\psi_u = (1-u)\epsilon + ua$ represents the linear interpolation between the source noise $\epsilon$ and the target action $a$ at flow time $u$. The effectiveness of $\mathcal{L}_{\text{past}}$ is empirically validated in our ablation study in Section 5.3. While self-imitation strategies have appeared in diffusion RL contexts (Yang et al., 2025), to the best of our knowledge, we are the first to uncover the fundamental role of Past Data CFM within the paradigm of RPG-driven flow policy optimization. Our analysis establishes that this regularization serves as a critical stabilizer for RPG-based flow optimization.

We introduce a second regularization term to explicitly promote exploration. Unlike prior Maximum Entropy approaches (Dong et al., 2025; Wang et al., 2024; Celik et al., 2025), which rely on approximating the entropy of the generative policy, we adopt a more direct strategy. Given the bounded action space in our setting, we propose to guide the flow policy toward uniformly sampled target actions via the CFM objective. While QVPO (Ding et al., 2024) explores a similar uniform-target regularization for diffusion policies, it employs state-dependent weighting. In contrast, we adopt a simple approach that treats all states encountered during rollouts equally. The proposed uniform exploration CFM objective, denoted as $\mathcal{L}_{\text{uni}}$, is defined as:

$$
\begin{aligned}
&\mathcal{L}_{\text{uni}}(\theta) \\
&= \mathbb{E}_{\substack{u \sim \mathcal{U}[0,1], \epsilon \sim p_0, \\ s \sim \mathcal{D}_{\text{rollout}}, a \sim p_{\text{uni}}}} \left[ \left\| v_\theta(\psi_u, u|s) - (a - \epsilon) \right\|^2 \right],
\end{aligned} \quad (12)
$$

where $\mathcal{D}_{\text{rollout}}$ denotes the set of states visited during the current iteration's rollout, and $p_{\text{uni}}$ represents the uniform distribution over the bounded action space. As before, $\psi_u$ denotes the linear interpolation. The empirical effectiveness of $\mathcal{L}_{\text{uni}}$ is analyzed in Section 5.3.

## 4.3. Value Function Training

We optimize the critic by minimizing the standard mean squared error loss (Xing et al., 2025; Xu et al., 2021):

$$
\mathcal{L}(\omega) = \mathbb{E} \left[ \left\| V_\omega(s) - y(s) \right\|^2 \right], \quad (13)
$$

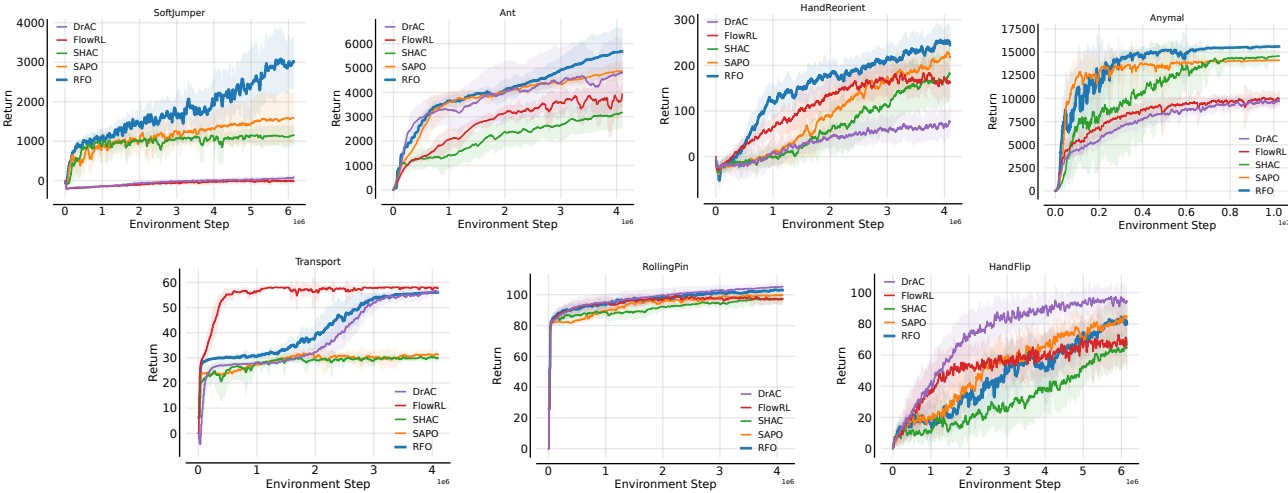

*Figure 3.* Training curves across seven tasks. Solid lines represent the mean return, while shaded regions indicate the standard deviation. Curves are smoothed using a 100-episode moving average.

where $y(s)$ denotes the TD-$\lambda$ target (Sutton & Barto, 2018). Following the dual-critic architecture in SAPO (Xing et al., 2025), we calculate the target value by averaging the predictions of two separate critic networks.

### 4.4. Overall Policy Objective and RFO Algorithm

Combining the RPG objective for the flow policy with the past data CFM regularization (Eq. (11)) and uniform exploration CFM regularization (Eq. (12)), RFO minimizes the total policy loss:

$$\mathcal{L}_{\text{policy}}(\theta) = -J(\theta) + c_{\text{past}}\mathcal{L}_{\text{past}}(\theta) + c_{\text{uni}}\mathcal{L}_{\text{uni}}(\theta), \quad (14)$$

where $c_{\text{past}}$ and $c_{\text{uni}}$ are hyperparameters weighting the stability and exploration regularization terms, respectively.

The overall RFO algorithm is described in Algorithm 1. We follow the SHAC paradigm, rolling out with flow policies to collect short-horizon trajectories. Then, we compute policy gradients and the gradients of the two CFM regularization terms. Combining and weighting these gradients, we update the policy. Afterwards, we update the value function by minimizing Eq. (13).

### 4.5. Action Chunking

Action chunking has proven beneficial for improving the temporal consistency of robot action generation (Black et al., 2025b), a critical factor in robotic applications. Therefore, we provide an extension to RFO that incorporates action chunking. In this variant, the flow policy predicts an action chunk of size $C$ based on the current observation, encompassing both current and future actions. Subsequently, the agent executes these actions for $C$ steps before replanning based on the new observation.

## 5. Experiments

We conduct experiments to answer the following questions: **(i)** How does RFO's performance compare to RPG-based approaches with Gaussian policies? **(ii)** How does RFO's performance compare to other RL algorithms for flow or diffusion-based approaches? **(iii)** Are the two regularization terms critical to RFO's performance? **(iv)** How does RFO work with the action chunking mechanism?

### 5.1. Experimental Setup

We evaluate our method on a comprehensive set of locomotion and manipulation tasks, utilizing either state or visual inputs within the differentiable simulators Rewarped (Xing et al., 2025) and DFlex (Xu et al., 2021; Georgiev et al., 2024).

**Tasks.** Our experiments span diverse rigid and soft-body dynamics, categorized into:

- Locomotion tasks: Ant, ANYmal (Hutter et al., 2017) (rigid quadruped), and Soft Jumper (visual soft body).

- Manipulation tasks: Hand Reorient (cube rotation), Rolling Pin (dough flattening), Hand Flip (object flipping), and Transport (liquid transport).

Visual inputs are used for Soft Jumper and the last three manipulation tasks.

**Baselines.** We compare RFO against strong baselines from two categories:

- RPG Baselines: SAPO (Xing et al., 2025) (SOTA RPG-based method with maximum entropy) and SHAC (Xu

*Table 1.* Final performance evaluation after training. Results are reported as mean $\pm$ standard deviation across all random seeds. Each seed is evaluated over 128 episodes. Normalized score $= \frac{\text{Mean Reward}}{\text{SHAC Mean Reward}}$.

(a) Raw Final Evaluation Rewards

| Task | DrAC | FlowRL | SHAC | SAPO | RFO (Ours) |
|---|---|---|---|---|---|
| Soft Jumper | $83.56 \pm 67.60$ | $-75.71 \pm 139.54$ | $1148.87 \pm 233.84$ | $1597.99 \pm 653.87$ | $\mathbf{3023.96 \pm 603.61}$ |
| Ant | $4846.41 \pm 1412.76$ | $4083.45 \pm 1082.79$ | $3143.08 \pm 569.50$ | $4865.36 \pm 363.77$ | $\mathbf{5677.88 \pm 964.49}$ |
| Hand Reorient | $73.74 \pm 34.22$ | $135.57 \pm 39.34$ | $174.63 \pm 57.54$ | $213.44 \pm 33.95$ | $\mathbf{258.84 \pm 21.80}$ |
| ANYmal | $9554.46 \pm 928.33$ | $9717.58 \pm 1619.35$ | $14568.97 \pm 652.72$ | $14095.90 \pm 82.02$ | $\mathbf{15622.76 \pm 200.87}$ |
| Transport | $57.07 \pm 0.85$ | $\mathbf{58.04 \pm 0.19}$ | $30.00 \pm 1.43$ | $31.35 \pm 1.24$ | $56.00 \pm 0.28$ |
| Rolling Pin | $\mathbf{104.71 \pm 2.00}$ | $92.49 \pm 2.88$ | $97.29 \pm 3.44$ | $99.79 \pm 2.42$ | $103.11 \pm 1.26$ |
| Hand Flip | $\mathbf{89.82 \pm 4.59}$ | $54.80 \pm 18.55$ | $65.57 \pm 11.75$ | $83.00 \pm 7.49$ | $82.65 \pm 11.65$ |

(b) SHAC-normalized Final Evaluation Rewards

| Task | DrAC | FlowRL | SHAC | SAPO | RFO (Ours) |
|---|---|---|---|---|---|
| Soft Jumper | $0.07 \pm 0.06$ | $-0.07 \pm 0.12$ | $1.00 \pm 0.20$ | $1.39 \pm 0.57$ | $\mathbf{2.63 \pm 0.53}$ |
| Ant | $1.54 \pm 0.45$ | $1.30 \pm 0.34$ | $1.00 \pm 0.18$ | $1.55 \pm 0.12$ | $\mathbf{1.81 \pm 0.31}$ |
| Hand Reorient | $0.42 \pm 0.20$ | $0.78 \pm 0.23$ | $1.00 \pm 0.33$ | $1.22 \pm 0.19$ | $\mathbf{1.48 \pm 0.12}$ |
| ANYmal | $0.66 \pm 0.06$ | $0.67 \pm 0.11$ | $1.00 \pm 0.04$ | $0.97 \pm 0.01$ | $\mathbf{1.07 \pm 0.01}$ |
| Transport | $1.90 \pm 0.03$ | $\mathbf{1.93 \pm 0.01}$ | $1.00 \pm 0.05$ | $1.05 \pm 0.04$ | $1.87 \pm 0.01$ |
| Rolling Pin | $\mathbf{1.08 \pm 0.02}$ | $0.95 \pm 0.03$ | $1.00 \pm 0.04$ | $1.03 \pm 0.02$ | $1.06 \pm 0.01$ |
| Hand Flip | $\mathbf{1.37 \pm 0.07}$ | $0.84 \pm 0.28$ | $1.00 \pm 0.18$ | $1.27 \pm 0.11$ | $1.26 \pm 0.18$ |
| **Average** | 1.01 | 0.91 | 1.00 | 1.21 | **1.60** |

et al., 2021) (short-horizon RPG).

- Flow/Diffusion RL Baselines: DrAC (Wang et al., 2025b) (a SOTA diffusion policy) and FlowRL (Lv et al., 2025) (a SOTA flow RL method).

All tasks are evaluated using at least 10 random seeds for each algorithm. Further experiment details are provided in Appendix B.

### 5.2. Main Experiment Results

We show the training curves in Figure 3 and final performance evaluation results in Table 1.

In the Soft Jumper, Ant, ANYmal, and Hand Reorient tasks, RFO outperforms all baselines by substantial margins. Most notably, in the Soft Jumper task, RFO achieves a final reward nearly $2\times$ that of the best baseline, underscoring its capability in high-dimensional control settings (action space $\mathbb{R}^{222}$). This superiority extends to rigid-body dynamics, where RFO consistently dominates strong baselines, securing performance gains of approximately 21% on Hand Reorient and 17% on Ant, while boosting the reward by over 1000 points on the ANYmal task. Furthermore, RFO demonstrates remarkable robustness on the Rolling Pin and Transport tasks, achieving performance comparable to the state-of-the-art baselines. As shown in the aggregate performance analysis (Table 1 (b)), which normalizes scores against the SHAC baseline, RFO delivers superior overall performance across the diverse task suite.

From the RPG perspective, RFO consistently improves upon SHAC and SAPO, validating the efficacy of integrating flow policies into the RPG framework. Compared to Flow and Diffusion baselines, RFO demonstrates a clear advantage by leveraging analytical gradients from RPG. This becomes particularly evident in the Soft Jumper, ANYmal, and Hand Reorient tasks, where DrAC and FlowRL struggle, while RFO demonstrates superior performance. Additionally, DrAC requires 20 denoising steps for action sampling, which not only slows down inference but also significantly increases training time compared to RFO, since it needs to backpropagate through all 20 action generation steps. Furthermore, despite being an on-policy algorithm, RFO achieves sample efficiency comparable to off-policy flow-based methods thanks to RPG, while avoiding the need to approximate intractable action log-likelihoods.

### 5.3. Ablation Study

In this section, we conduct ablation experiments to answer the following questions: **(i)** Are the two proposed regularization terms, past data CFM regularization and uniform exploration CFM regularization, critical for RFO's performance? **(ii)** How does RFO perform under a range of hyperparameters, including the weights for the two CFM objectives and the number of flow integration steps?

To answer the first question, we conduct experiments on the Ant, Soft Jumper, and Transport tasks. We compare the full RFO algorithm against two variants: RFO without past data

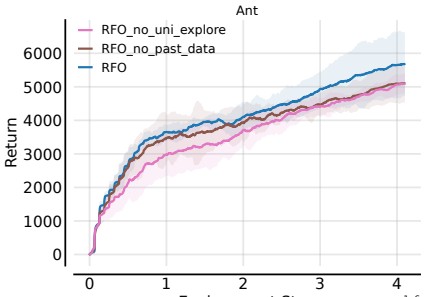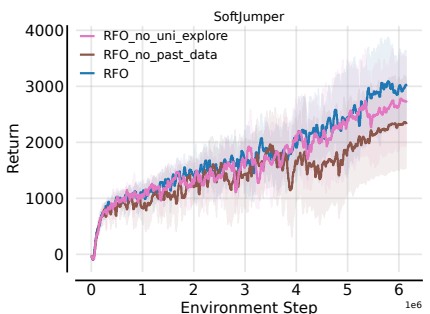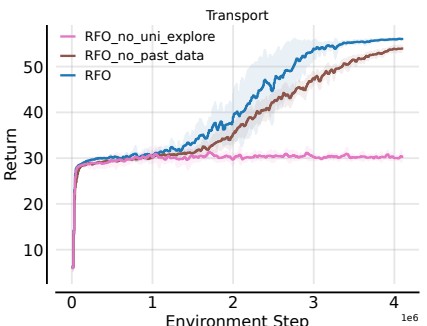

*Figure 4.* Ablation study for the effectiveness of past data CFM regularization and uniform exploration CFM regularization. The results show that the two proposed regularization terms are critical for RFO's performance.

CFM regularization and RFO without uniform exploration CFM regularization. The training curves are presented in Figure 4.

Without past data CFM regularization, RFO's performance significantly degrades across all three tasks, suggesting that this term is crucial for both training stability and final performance. Similarly, removing the uniform exploration regularization leads to performance drops across all tasks. Most notably, on the Transport task, omitting uniform exploration regularization causes RFO's performance to degrade to the levels of SHAC and SAPO, clearly highlighting the effectiveness of the uniform random exploration strategy.

To answer the second question, we conduct experiments on the Soft Jumper task. We evaluate RFO with different combinations of $c_{past}$ and $c_{uni}$, which control the weights of the two regularization terms. We also evaluate RFO with varying numbers of flow Euler integration steps. The training curves are provided in Appendix C, demonstrating that RFO exhibits robustness across a wide range of hyperparameters.

### 5.4. Action Chunking Results

Figure 6 in Appendix D illustrates the training curves for RFO with action chunking across the evaluated tasks. This result is particularly encouraging given the inherent challenge: the flow policy must predict action sequences for future time steps based solely on the current observation, without the guidance of expert data used in imitation learning. Such optimization is harder, as the policy must generate valid future actions without access to the corresponding intermediate states. Consequently, this setup suggests that integrating offline-to-online procedures could be a promising direction to further enhance RPG-based flow policies.

### 5.5. Memory Usage and Training Wall-Clock Time

We report the computational cost of RFO on the Soft Jumper task, which uses pixel inputs and is therefore representative of the more demanding settings in our benchmark. All experiments are conducted on a machine with 4 NVIDIA A40 GPUs, with one GPU dedicated to each seed. We run 4 seeds for RFO as well as the baselines SAPO (Xing et al., 2025), FlowRL (Lv et al., 2025), and DrAC (Wang et al., 2025b) for reference. Table 2 summarizes inference time per action, training throughput (environment steps per second), wall-clock training time, and GPU memory usage, reported as mean [min, max] across the four seeds.

**Training Time.** RFO trains for 2.68 hours on this task, comparable to the Gaussian-based RPG baseline SAPO (2.36 hours), with only 0.32 hour of additional overhead. This indicates that replacing the Gaussian policy with a flow policy under the RPG framework introduces only a modest training-time cost.

**Memory.** RFO uses 7.10 GB of GPU memory, 14.8% of an A40's 48 GB capacity. We consider this a reasonable cost given RFO's algorithmic performance.

**Inference Latency.** RFO requires 2.83 ms per action with 4-step flow integration, suitable for real-time control applications.

## 6. Conclusion

In this work, we introduce Reparameterization Flow Policy Optimization, a novel framework that bridges the gap between flow-based generative policies and the high sample efficiency of Reparameterization Policy Gradients. By integrating tailored regularization terms for training stability and exploration, RFO achieves strong performance. Empirical evaluations across diverse rigid and soft-body control tasks demonstrate that RFO achieves SOTA performance. We further explored an action-chunking RFO variant. Future directions include extending RFO to offline-to-online reinforcement learning settings.

*Table 2.* Computational efficiency on the Soft Jumper task. Values are reported as mean [min, max] across 4 seeds.

| Method | Inference Time (ms) | Training Throughput (steps/sec) | Training Time (h) | GPU Memory (GB) |
| --- | --- | --- | --- | --- |
| SAPO | 0.97 [0.94, 1.21] | 727 [710, 743] | 2.36 [2.31, 2.40] | 4.23 |
| FlowRL | 1.22 [1.19, 1.40] | 56 [56, 57] | 30.41 [30.26, 30.60] | 1.50 |
| **RFO (Ours)** | 2.83 [2.74, 2.96] | 635 [629, 642] | 2.68 [2.66, 2.70] | 7.10 |
| DrAC | 8.73 [8.39, 9.50] | 107 [105, 108] | 16.04 [15.89, 16.20] | 2.33 |

## Impact Statement

This paper presents work whose goal is to advance the field of Machine Learning. There are many potential societal consequences of our work, none of which we feel must be specifically highlighted here.

## Acknowledgements

This work was supported by the National Natural Science Foundation of China Grant 52494974.

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

# A. Task Details

We conduct experiments with two differentiable physics simulators: DFlex (Xu et al., 2021; Georgiev et al., 2024) from NVIDIA and Rewarped (Xing et al., 2025). More specifically, we use AHAC's version of the DFlex simulator and Rewarped version 1.3.0 from `https://github.com/rewarped/rewarped`.

Here, we provide details of the tasks used in the main experiments.

### A.1. Ant

Ant is a classical locomotion task using the Rewarped simulator (Xing et al., 2025), where the goal is to maximize forward velocity. Ant uses state inputs. The state space is $R^{37}$ and the action space is $R^8$, including joint torques (Xing et al., 2025).

### A.2. Hand Reorient

Hand Reorient is a task from Rewarped (Xing et al., 2025), where the goal is to reorient a cube with an Allegro dexterous hand, using state inputs. The state space is $R^{72}$ and the action space is $R^{16}$, including joint torques (Xing et al., 2025).

### A.3. Rolling Pin

Rolling Pin is a manipulation task from Rewarped (Xing et al., 2025), where the goal is to flatten dough with a rolling pin, using pixel inputs. The observation space consists of $\left[R^{250 \times 3}, R^3, R^3\right]$, and the action space is $R^3$, including relative position and orientation (Xing et al., 2025).

### A.4. Soft Jumper

Soft Jumper is a soft-body locomotion task from Rewarped (Xing et al., 2025) with pixel inputs. The observation space consists of $\left[R^{204 \times 3}, R^3, R^3, R^{222}\right]$, and the action space is $R^{222}$, including tetrahedral activations (Xing et al., 2025).

### A.5. Hand Flip

Hand Flip is a soft-object manipulation task from Rewarped (Xing et al., 2025) using a Shadow Hand and pixel inputs. The observation space consists of $\left[R^{250 \times 3}, R^3, R^{24}\right]$, and the action space is $R^{24}$, including relative position and orientation (Xing et al., 2025).

### A.6. Transport

Transport is a soft-object manipulation task from Rewarped (Xing et al., 2025) with pixel inputs. The observation space consists of $\left[R^{250 \times 3}, R^3, R^3, R^3\right]$, and the action space is $R^3$, including relative position (Xing et al., 2025).

### A.7. ANYmal

ANYmal is a locomotion task from DFlex (Georgiev et al., 2024) with state inputs, where the goal is to maximize the forward velocity of a quadruped robot (Hutter et al., 2017). The state space is $R^{49}$ and the action space is $R^{12}$.

# B. Baselines, Implementation Details and Hyperparameters

### B.1. Baselines and Hyperparameters

We benchmark RFO against four baselines: SHAC (Xu et al., 2021), SAPO (Xing et al., 2025), DrAC (Wang et al., 2025b), and FlowRL (Lv et al., 2025). For SAPO, we utilize the official repository[1], from which we also adopt the improved SHAC implementation as it demonstrates improved and stable performance. For DrAC (Diffusion Policy) and FlowRL (Flow Policy), we adapt their official implementations to support parallelized environments and incorporate visual encoders for pixel-based inputs. We use their default sampler and integration steps for action sampling.

We tune the hyperparameters of DrAC and FlowRL individually for each task, finding that the Update-to-Data (UTD) ratio

---

[1] `https://github.com/etaoxing/mineral`

affects their performance significantly (FlowRL defaults to $0.5$, as in their official repository). Note that tuning DrAC with UTD ratios $\geq 1$ is computationally prohibitive, as training would require weeks to complete. Furthermore, we observe that a higher UTD ratio does not necessarily translate to better performance. We align the neural network sizes across all algorithms. For visual input tasks, we use DP3 Point Net with layers [64, 128, 256] and 64-dimension output, average pooling for all algorithms. Shared hyperparameters are detailed in Table 3.

*Table 3.* Common hyperparameters for all algorithms.

| | *shared* | RFO | SHAC | SAPO | DrAC | FlowRL |
|---|---|---|---|---|---|---|
| Horizon $H$ | 32 | | | | | |
| Epochs for critics $L$ | | 16 | 16 | 16 | see UTD ratio | see UTD ratio |
| Epochs for actors $M$ | | 1 | 1 | 1 | see UTD ratio | see UTD ratio |
| Discount $\gamma$ | 0.99 | | | | | |
| TD/GAE $\lambda$ | 0.95 | | | | N.A. | N.A. |
| Actor MLP | $(400, 200, 100)$ | | | | | |
| Critic MLP | $(400, 200, 100)$ | | | | | |
| Actor $\eta$ | | $2e-3$ | $2e-3$ | $2e-3$ | $3e-4$ | $3e-4$ |
| Critic $\eta$ | $5e-4$ | | | | $3e-4$ | $3e-4$ |
| Learning rate schedule | - | linear | linear | linear | N.A. | N.A. |
| Optim type | AdamW | | | | | |
| Optim $(\beta_1, \beta_2)$ | $(0.9, 0.999)$ | | | | | |
| Norm type | LayerNorm | | | | | |
| Activation type | SiLU | | | | | |

*Table 4.* The number of parallel environments used for each environment. These values are kept the same as in the official implementations: we follow the AHAC repository (https://github.com/imgeorgiev/DiffRL) for the DFlex tasks and the Rewarped repository.

| | Ant | Transport | Hand Flip | Rolling Pin | Soft Jumper | Anymal | Hand Reorient |
|---|---|---|---|---|---|---|---|
| Num Envs | 64 | 32 | 32 | 32 | 32 | 128 | 64 |

*Table 5.* UTD ratios for FlowRL and DrAC.

| | Ant | Transport | Hand Flip | Rolling Pin | Soft Jumper | Anymal | Hand Reorient |
|---|---|---|---|---|---|---|---|
| DrAC | 0.5 | 0.125 | 0.5 | 0.25 | 0.125 | 0.25 | 0.125 |
| FlowRL | 0.125 | 0.125 | 0.25 | 0.5 | 0.5 | 0.25 | 0.25 |

**Number of Seeds and Runs.** The SHAC and SAPO results on Hand Reorient and ANYmal are taken from our prior work. On Hand Reorient, all methods (RFO, DrAC, FlowRL, SHAC, SAPO) use 12 random seeds with 2 runs per seed (24 runs total per algorithm) to account for simulator stochasticity, matching the configuration of our prior work. On ANYmal, due to computational constraints, RFO, DrAC, and FlowRL use 10 seeds with a single run, while SHAC and SAPO retain the 12-seed $\times$ 2-run configuration. Despite this asymmetry, the observed performance differences are sufficiently large to support our conclusions. For the remaining tasks, all algorithms are evaluated with 10 random seeds.

$n$-**step TD Augmentation for DrAC and FlowRL on ANYmal.** We observe that the original DrAC and FlowRL implementations struggle on the ANYmal task. To enable a stronger baseline comparison, we additionally augment them with an $n$-step TD target on the ANYmal task, which is not present in their official repositories. We tune $n$ and adopt $n = 10$.

## B.2. Implementation Details of RFO

Since the flow velocity field can generate unbounded actions, we apply a tanh squashing function to the output. We utilize the pre-tanh values as targets for both past data CFM regularization and uniform CFM regularization (although we found that using tanh-transformed actions as targets is also effective). Additionally, we observe that excessively large pre-tanh targets can cause issues for the CFM loss; hence, we apply clamping to bound them. We clamp pre-tanh targets to a maximum absolute value of 1 for all tasks, except for HandFlip and RollingPin where we clamp to 5.

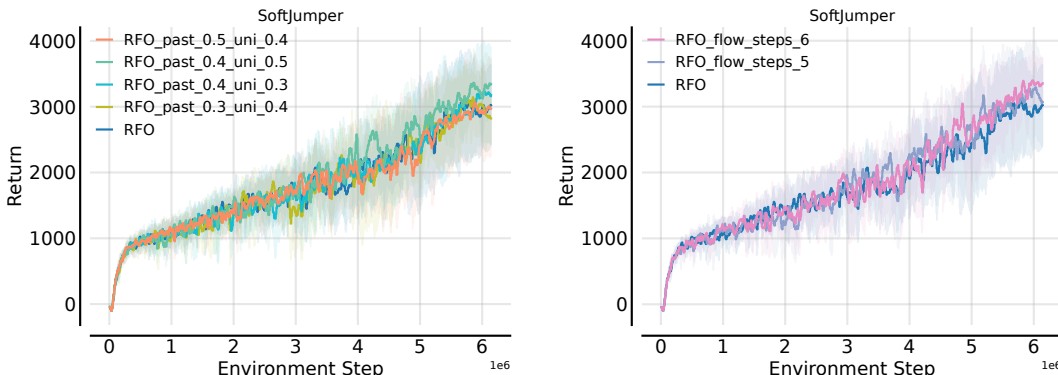

*Figure 5.* (a) Ablation study on different weight combinations for past data CFM regularization and uniform exploration CFM regularization. (b) Ablation study on the number of flow integration steps.

Detailed unique hyperparameters for RFO are shown in Table 6.

*Table 6.* RFO's unique hyperparameters.

|  | Ant | Transport | Hand Flip | Rolling Pin | Soft Jumper | Anymal | Hand Reorient |
|---|---|---|---|---|---|---|---|
| $c_{\text{past}}$ | 0.2 | 0.1 | 5e-4 | 5e-3 | 0.4 | 0.1 | 0.2 |
| $c_{\text{uni}}$ | 0.2 | 0.1 | 0.2 | 5e-3 | 0.4 | 0.1 | 0.01 |

## C. Sensitivity to Hyperparameters

In this section, we conduct ablation experiments on: (i) different weight combinations for past data CFM regularization and uniform exploration CFM regularization, and (ii) the number of flow integration steps (RFO uses 4 flow steps across all tasks by default). As shown in Figure 5, RFO demonstrates robust performance across a wide range of hyperparameters.

## D. RFO with Action Chunking Results

Figure 6 illustrates the training curves for RFO augmented with the action chunking mechanism. We observe that RFO with action chunking generally maintains strong performance, largely matching that of the standard RFO baseline (without chunking) across the majority of evaluated environments. Although a minor performance degradation is observed in certain tasks, the results remain competitive. This is expected, as the flow policy is required to predict a sequence of actions simultaneously, which increases the complexity of policy optimization.

## E. Details of KL Divergence Estimation

In this section, we detail the approximation of the KL divergence between consecutive policy updates on the Soft Jumper task. Following the methodology in FPO (McAllister et al., 2025), for a given state-action pair $(s, a)$ and flow policies $\pi_\theta$ and $\pi_{\theta'}$, we can approximate the likelihood ratio $\frac{\pi_\theta(a|s)}{\pi_{\theta'}(a|s)}$ using the difference in their Conditional Flow Matching (CFM) losses. Specifically, this ratio is approximated as $\exp(L_{\theta'}^{\text{CFM}}(a|s) - L_\theta^{\text{CFM}}(a|s))$. Leveraging this relationship, we sample actions from the old/behavior policy $\pi_\theta$ and compute the CFM losses for both $\pi_\theta$ and $\pi_{\theta'}$ using these samples as targets. Finally, we employ the K3 estimator (Schulman, 2020) to estimate the KL divergence $\text{KL}(\pi_\theta \| \pi_{\theta'})$.

## F. Design Choice

In this section, we justify our decision to include rollout states and actions from both the current and previous iterations in $\mathcal{D}_{\text{recent}}$. As shown in Figure 7, relying solely on the current iteration results in significantly slower learning. Conversely, using data from three iterations leads to degraded performance and reduced learning speed. In contrast, we find that including state-action pairs from both the current and previous iterations yields the best results in terms of final performance and

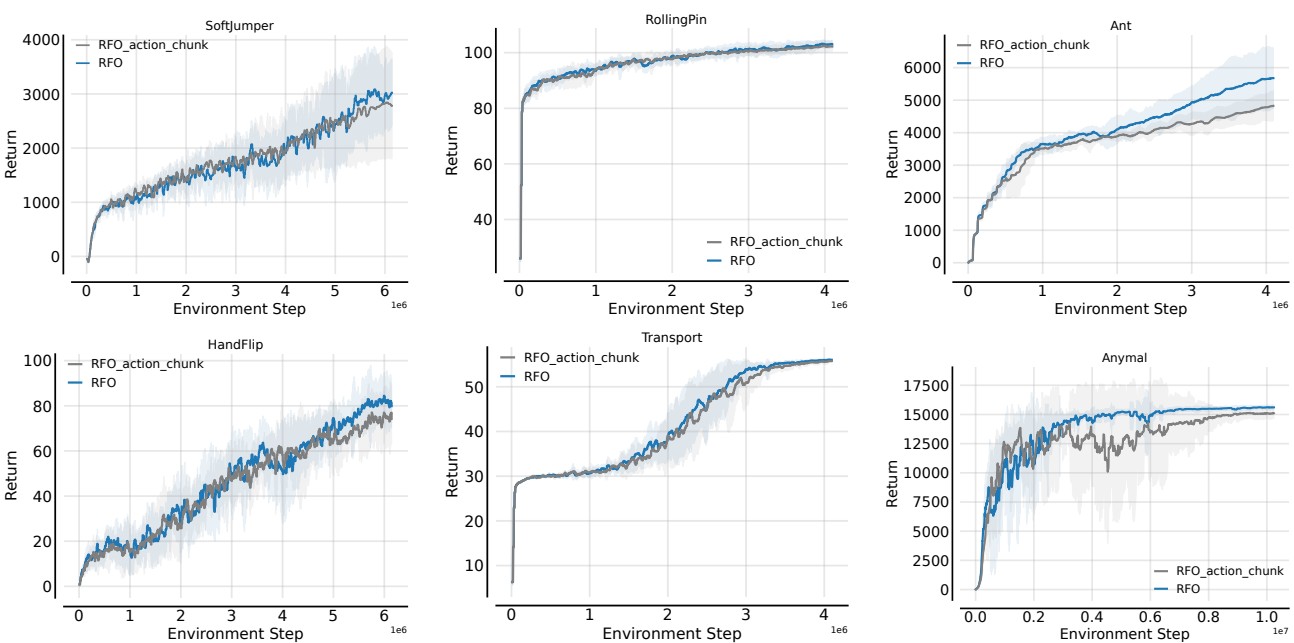

*Figure 6.* Training curves of RFO with the action chunking mechanism.

convergence speed.

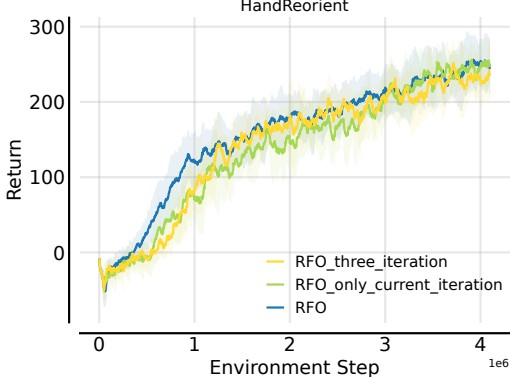

*Figure 7.* Comparison of $\mathcal{D}_{\text{recent}}$ including data from: the current iteration only, the current + previous iterations, and three iterations of rollout data.

