# OpenReview forum: "Reparameterization Flow Policy Optimization"
_ICML.cc/2026/Conference — ICML 2026 regular_

### Official Review · Reviewer_n6Wa · 2026-03-07

**Soundness:** 3
**Presentation:** 2
**Significance:** 2
**Originality:** 2
**Overall Recommendation:** 4
**Confidence:** 3

**Summary:**

In this paper, the authors investigate reparameterisation policy gradient (RPG) for flow-based policies.
The paper builds on recent work on reparameterisation policy gradient and flow-based policies.
Their key insight is that the ODE solver is a differentiable mapping from a source noise to actions.
While a seemingly obvious insight, to the best of my knowledge, this is the first work combining flow matching and RPG.
They then show that they can train flow policies using RPG to get high sample efficiency without approximating intractable likelihoods.
They also introduce loss terms to promote stability and exploration.
They test their method in differentiable simulators (Rewarped and DFlex) for both rigid and soft bodies and state and visual inputs.

**Compliance With Llm Reviewing Policy:**

Affirmed.

**Final Justification:**

Overall, I lean towards acceptance, but I can see why other reviewers might not, and I would not champion the paper in that case.

The aggregate performance metrics and confidence intervals have significantly improved the statistical significance of the results and it is nice to see how RFO performs in terms of compute/memory. In light of the changes, I increased my score from 3 to 4.

**Key Questions For Authors:**

1. What led you to the uniform exploration loss in Equation 13? Please can you provide some more intuition for why this is a good way to promote exploration?
2. Backprop through the flow matching solver sounds like it will eat VRAM. How much VRAM did backprop through your models use? What GPUs did you train on? Did you need to use activation checkpointing?
    - How do other methods perform if given the same amount of compute?

**Limitations:**

Limitations do not appear to have been discussed.

**Strengths And Weaknesses:**

Overall, I found the paper well written and easy to follow.
Whilst the work is rather incremental, it is an important direction which is of interest to the community.
I also liked that they tested their method with both state and pixel-based observations and for soft and rigid bodies.

My main concern is with the experiments and results.
Figure 3 reports training curves in 7 tasks.
I do not think 7 tasks are enough, and I would encourage the authors to test in more environments.
Second, I would like to see aggregate performance metrics,  see rliable for how to report results for RL benchmarks with a handful of runs: https://github.com/google-research/rliable.
For example, it would be nice to aggregate normalised returns for all environments and then show a training curve where the y-axis is the inter-quartile mean (IQM) with stratified bootstrap confidence intervals.
Moreover, Table 1b reports the average over all environments.
Again, it would be nice if the authors could follow advice from rliable, and report IQM, median, etc.


# Minor corrections
- In Section 3.1, the transition dynamics $p(s' \mid s, a)$ are stochastic but the reward model $r(s,a)$ is deterministic. This feel a bit odd. Surely the reward needs to depend on $s'$ to be deterministic, i.e. $r(s,a,s')$?
- The text in Figures 2 and 3 is too small and should be made larger.

---

> ### Author Rebuttal · Authors · 2026-03-30
>
> Thanks for your constructive comments!
>
> #### **W1: Tasks, IQM, Median**
>
> We have added aggregate performance metrics following rliable guidelines. Our aggregate training curve (IQM) is at this link: [Aggregate Curve](https://anonymous.4open.science/r/greatday/aggregate_iqm_training_curve_preview.png). We report IQM and Median normalized by SHAC's **mean performance** per task, with 95% stratified bootstrap CIs:
>
> | Task | Metric | RFO (Ours) | SHAC | SAPO | DrAC | FlowRL |
> |:-----|:------:|:----------:|:----:|:----:|:----:|:------:|
> | SoftJumper | IQM | 2.66 | 0.99 | 1.26 | 0.07 | -0.07 |
> | SoftJumper | Median | 2.69 | 0.98 | 1.08 | 0.07 | -0.06 |
> | Ant | IQM | 1.73 | 1.01 | 1.54 | 1.63 | 1.38 |
> | Ant | Median | 1.70 | 1.02 | 1.54 | 1.62 | 1.38 |
> | HandReorient | IQM | 1.46 | 1.03 | 1.22 | 0.40 | 0.75 |
> | HandReorient | Median | 1.45 | 1.07 | 1.23 | 0.41 | 0.75 |
> | Anymal | IQM | 1.07 | 1.01 | 0.97 | 0.66 | 0.70 |
> | Anymal | Median | 1.07 | 1.01 | 0.97 | 0.68 | 0.68 |
> | Transport | IQM | 1.87 | 1.01 | 1.04 | 1.91 | 1.94 |
> | Transport | Median | 1.87 | 1.02 | 1.03 | 1.91 | 1.94 |
> | RollingPin | IQM | 1.06 | 1.01 | 1.03 | 1.08 | 0.95 |
> | RollingPin | Median | 1.06 | 1.01 | 1.03 | 1.08 | 0.96 |
> | HandFlip | IQM | 1.29 | 0.98 | 1.28 | 1.37 | 0.85 |
> | HandFlip | Median | 1.35 | 0.98 | 1.24 | 1.37 | 0.90 |
> | Average | IQM | 1.59 | 1.01 | 1.19 | 1.02 | 0.93 |
> | Average | Median | 1.60 | 1.01 | 1.16 | 1.02 | 0.94 |
>
> Aggregate 95% Stratified Bootstrap CI:
>
> | | IQM (95% CI) | Median (95% CI) |
> |---|:---:|:---:|
> | RFO (Ours) | 1.59 (1.53, 1.65) | 1.60 (1.52, 1.66) |
> | SHAC | 1.01 (0.97, 1.04) | 1.01 (0.97, 1.04) |
> | SAPO | 1.19 (1.14, 1.26) | 1.16 (1.13, 1.27) |
> | DrAC | 1.02 (0.97, 1.05) | 1.02 (0.98, 1.06) |
> | FlowRL | 0.93 (0.88, 0.97) | 0.94 (0.88, 0.97) |
>
> RFO achieves the highest aggregate performance across both IQM and Median, with non-overlapping 95% CIs against all baselines.
>
> Regarding task coverage: our 7 tasks are diverse and representative, spanning locomotion, soft-body control, dexterous manipulation, covering state-based and visual inputs, rigid and soft body dynamics. This coverage is comparable to or broader than prior RPG works (e.g., SHAC, SAPO).
>
> #### **Q1: Entropy**
>
> **Intuition:** In RPG, policies can quickly collapse to narrow action regions due to the aggressive nature of reparameterization gradients. To counteract this, we need an exploration mechanism. However, flow policies lack a tractable log-likelihood, making standard entropy bonuses inapplicable. CFM loss naturally measures the "distance" between two distributions. By choosing the uniform distribution over $[-1,1]^d$ as the target, which is the maximum entropy distribution on bounded action spaces, minimizing $L_{\text{uni}}$ effectively maximizes policy entropy without requiring explicit density evaluation.
>
> **Theoretical support:** As shown in FPO, the CFM loss serves as a tractable approximation to $D_{\text{KL}}(\pi_\theta \| \text{Uniform})$. Since $H(\pi_\theta) = \text{const} - D_{\text{KL}}(\pi_\theta \| \text{Uniform})$ on bounded support, minimizing this KL is equivalent to maximizing entropy.
>
> **Empirical support:** In the ablation on the Transport task, RFO without $L_{\text{uni}}$ exhibits a significantly higher random action CFM loss (2.19 vs. 1.09), confirming that $L_{\text{uni}}$ effectively pushes the policy closer to uniform. The performance difference (56 vs 30) clearly shows the benefits of $L_{\text{uni}}$.
>
> #### **Q2: Memory**
>
> To rigorously and fairly investigate the overhead, we conducted new experiments on a machine with 4 A40 GPUs. We ran 4 seeds each for RFO, DrAC, FlowRL, and SAPO on the SoftJumper task (visual input), each seed using one GPU. Results (mean [min, max]):
>
> | Method | Inference (ms) | Throughput (steps/s) | Time (h) | Memory (GB) |
> |:---|:---:|:---:|:---:|:---:|
> | SAPO | 0.97 [0.94, 1.21] | 727 [710, 743] | 2.36 [2.31, 2.40] | 4.23 |
> | FlowRL | 1.22 [1.19, 1.40] | 56 [56, 57] | 30.41 [30.26, 30.60] | 1.50 |
> | RFO (Ours) | 2.83 [2.74, 2.96] | 635 [629, 642] | 2.68 [2.66, 2.70] | 7.10 |
> | DrAC | 8.73 [8.39, 9.50] | 107 [105, 108] | 16.04 [15.89, 16.20] | 2.33 |
>
> Among generative methods, RFO is 11x faster than FlowRL and 6x faster than DrAC in training, comparable to Gaussian SAPO (2.68h vs 2.36h). RFO uses 7.10 GB (14.8% of A40's 48GB), a reasonable trade-off for its performance.
>
> We also let SAPO run for 10M env steps on SoftJumper, resulting in more training time than RFO. SAPO gets 1502.66, still worse than RFO.
>
> #### **Minor**
>
> Thanks for these. We consider deterministic dynamics, hence this point is fine. We will make figures larger in the revised version.
>
>
> ----
> We hope these clarifications address your concerns. If so, we wonder if you could kindly consider raising your score? We are happy to answer any further questions you may have. Thank you very much!

---

> > ### Author Rebuttal · Reviewer_n6Wa · 2026-04-03
> >
> > I thank the authors for their rebuttal.
> >
> > The rebuttal has addressed most of my concerns. The aggregate performance metrics and confidence intervals significantly improve the statistical significance of the results and it is nice to see how RFO performs in terms of compute/memory. In light of the changes, I will increase my score from 3 to 4.

---

> > > ### Author Response · Authors · 2026-04-03
> > >
> > > We are very grateful for your constructive suggestions and thanks for the reference of rliable. This is very helpful. We are very happy to receive your recognition. Thanks so much!

---

### Official Review · Reviewer_Uvi9 · 2026-03-11

**Soundness:** 3
**Presentation:** 3
**Significance:** 3
**Originality:** 3
**Overall Recommendation:** 4
**Confidence:** 3

**Summary:**

The paper proposes RFO, an on-policy RL algorithm that combines flow matching policies with reparameterization policy gradients (RPG). The key observation is that the Euler-discretized ODE integration in flow policies defines a differentiable map from noise to actions, which is directly compatible with RPG's requirement for a differentiable reparameterization. Two CFM-based regularization terms are introduced for stability and exploration.

**Compliance With Llm Reviewing Policy:**

Affirmed.

**Final Justification:**

I maintain my score. This is a qualified work

**Key Questions For Authors:**

1. Can you provide a controlled experiment where you add the past data CFM and uniform CFM regularization to SHAC (with Gaussian policy) to isolate the effect of the flow policy class?
2. The paper notes that DrAC requires 20 denoising steps (vs. RFO's 4 Euler steps), but no wall-clock training time comparisons are provided. Since RFO requires backpropagating through both the flow ODE steps and the environment dynamics, the memory and compute costs could be substantial. How does GPU memory scale with K (flow steps) × H (horizon)?
3. Do the learned flow policies exhibit actual multimodality on any of the tasks?

**Limitations:**

yes

**Strengths And Weaknesses:**

## Strengths
-  The idea of avoiding likelihood computation by leveraging RPG is simple but useful.
- The ablation study in Section 5.3 clearly demonstrates that both regularization terms are necessary.
- RFO avoids the intractable log-likelihood problem of on-policy flow RL methods (DPPO, Reinflow, FPO, GenPO) while being compatible with parallel differentiable simulators.

## Weaknesses

- There is no formal analysis of how L_uni relates to the actual entropy of the induced policy distribution. Is there any evidence that L_uni actually increases policy entropy?

-  RFO outperforms SHAC/SAPO (Gaussian RPG) on most tasks, but it is unclear why. On tasks like Ant, the improvement over SAPO could simply be due to the regularization terms rather than the expressiveness of the flow model. An ablation experiment adding the same regularization terms to SHAC/SAPO would help.

-  FPO, as a common baseline for on-policy flow RL method, should be added to the experimental comparison.


## Minor Issues

- The notation inconsistency between ε_t and x_0 for the source noise.
- Eq. (10) has an extra closing parenthesis.

---

> ### Author Rebuttal · Authors · 2026-03-30
>
> Thanks for your constructive comments!
>
> #### **W1: Entropy**
>
> We provide both theoretical and empirical justifications.
>
> Theoretically, the uniform distribution over $[-1,1]^d$ is the maximum entropy distribution on this bounded support. As shown in FPO, the CFM loss serves as a tractable approximation to the KL divergence between distributions. Therefore, minimizing $L_{\text{uni}} \approx D_{\text{KL}}(\pi_\theta \| \text{Uniform})$ is approximately maximizing the policy entropy $H(\pi_\theta)$ up to a constant.
>
> Empirically, we monitor the random action CFM loss as a proxy for the distance between the policy and the uniform distribution. In the ablation on the Transport task,  RFO without $L_{\text{uni}}$ exhibits a significantly higher random action CFM loss (2.19 vs. 1.09), confirming that $L_{\text{uni}}$ effectively pushes the policy distribution closer to uniform, thereby increasing entropy. The performance differences (56 vs 30) are clear on Transport, showing the benefits of $L_{\text{uni}}$.
>
> #### **W2, Q1: Controlled Experiments, Regularizations Applied to SHAC**
>
> Thanks! This is a great question. To investigate this, we conduct a new ablation experiment on the SoftJumper task. SHAC, with Gaussian policies, does not have a velocity field, hence CFM does not directly apply. Instead, we apply BC loss to the same RFO's $D_{recent}$ buffer and actions from uniform distributions. The results are summarized in the following table and it shows that these regularizations do not help the performance of SHAC.
>
> | | SHAC | SHAC w/ Reg. | RFO |
> |:---:|:----:|:------------:|:---:|
> | SoftJumper  | 1148.87 ± 233.84 | 1093.06 ± 325.10 | 3023.96 ± 603.61 |
>
> #### **W3: FPO**
>
> We re-implemented FPO in PyTorch within our unified framework to ensure a fair comparison, as the original FPO codebase uses JAX/Brax for continuous control tasks, which is incompatible with DFlex/Rewarped (they run with PyTorch). We aligned the neural network sizes, optimizer, etc. We did our best to tune FPO's hyperparameters, including clipping bounds, learning rates, update epochs, explorations, etc within this limited time window and compute budget. The results show that RFO outperforms FPO on SoftJumper and Ant tasks.
>
> | | RFO | FPO |
> |:---:|:----:|:---:|
> | SoftJumper | 3023.96 ± 603.61  |   -139.13 ± 51.82|
> | Ant | 5677.88 ± 964.49 |  2516.19 ± 1215.05|
>
> #### **Q2: Memory**
>
> We ran 4 seeds each for RFO (Ours), DrAC, FlowRL, and SAPO on the SoftJumper task, with each seed using a single NVIDIA A40 GPU.
>
> The results, including GPU memory usage, wall-clock training time, policy inference time, and training throughput, are summarized below (reported as **mean [min, max]** across seeds):
>
> | Method | Inference Time (ms) | Training Throughput (steps/sec) | Training Time (h) | Memory (GB) |
> | :--- | :--- | :--- | :--- | :--- |
> | SAPO | 0.97 [0.94, 1.21] | 727 [710, 743] | 2.36 [2.31, 2.40] | 4.23 |
> | FlowRL | 1.22 [1.19, 1.40] | 56 [56, 57] | 30.41 [30.26, 30.60] | 1.50 |
> | RFO  | 2.83 [2.74, 2.96] | 635 [629, 642] | 2.68 [2.66, 2.70] | 7.10 |
> | DrAC | 8.73 [8.39, 9.50] | 107 [105, 108] | 16.04 [15.89, 16.20] | 2.33 |
>
> RFO is the fastest generative-model-based method in training (11x faster than FlowRL, 6x faster than DrAC) and comparable to the non-generative SAPO. For inference, RFO (2.83 ms, 4 Euler steps) is 3.1x faster than DrAC (8.73 ms, 20 DDPM steps).
>
> Regarding memory scaling: (1) in our ablations varying the number of flow steps K from 4 to 10, we observe no significant change in GPU memory, as the per-step flow ODE overhead is small relative to the environment dynamics rollout. This range of K is sufficient for all tasks in our experiments. (2) GPU memory scales approximately linearly with the rollout horizon H, but this is inherent to backpropagation through differentiable dynamics in all RPG-based methods.
>
> #### **Q3: Multimodal**
>
> While the flow policy is architecturally capable of representing multimodal distributions, in practice we observe that each trained policy converges to a single behavioral mode. However, the expressive generative structure enables RFO to find high-reward strategies that Gaussians do not find. A concrete example is the **SoftJumper** task: RFO discovers **specialized high-jumping behaviors** (COM height reaching 5.7× body height), whereas SAPO (Gaussian) does not find this mode. A snapshot of this behavior is available at this link: [JumpingBehavior](https://anonymous.4open.science/r/greatday/JumpingBehavior.png). RFO also converges to fast forward running (1.9× body height, avg speed 2.57) in other seeds. This behavioral diversity suggests that the flow policy's rich representational capacity facilitates exploration, enabling discovery of high-reward strategy that unimodal Gaussian policies are unlikely to reach.
>
> ----
> We hope these clarifications address your concerns. If so, we wonder if you could kindly consider raising your score? We are happy to answer any further questions you may have.

---

> > ### Author Rebuttal · Reviewer_Uvi9 · 2026-04-01
> >
> > I don't quite understand the message you try to convey through the JumpingBehavior link.

---

> > > ### Author Response · Authors · 2026-04-02
> > >
> > > We are sorry for the confusion. First, we provide a complete recording of the jumping strategy for RFO, as shown in this [jumpervideo](https://anonymous.4open.science/r/greatday/jumper.mp4).
> > >
> > > SoftJumper is a challenging soft-body control task with an action dimension of 222. The reward function is defined as:
> > >
> > > **reward = 10.0 × forward velocity + 30.0 × relative height + 0.001 × action norm**
> > >
> > > This reward function encourages high-jumping strategies, as the height coefficient is 3× larger than that of forward velocity.
> > >
> > > Through this video, we want to express two points:
> > >
> > > (i) **Thanks to the flow policy, RFO discovers strategies that SAPO does not find.** As shown in the video, the RFO agent finds a high-jumping strategy enabled by the flow policy, whereas Gaussian policies (like SAPO) fail to find this mode (they tend to run faster instead). This is also why RFO achieves a higher reward than SAPO on the SoftJumper task.
> > >
> > > (ii) **Although each seed of RFO converges to a single behavioral mode, different seeds discover a diverse set of strategies.** For example, RFO also converges to a fast-forward-running mode in some seeds.
> > >
> > >
> > > ----
> > > We hope this clarification address your concern. If so, we wonder if you could kindly consider raising your score? Thanks!

---

### Official Review · Reviewer_Eu4g · 2026-03-11

**Soundness:** 3
**Presentation:** 3
**Significance:** 3
**Originality:** 2
**Overall Recommendation:** 4
**Confidence:** 2

**Summary:**

RFO combines Flow Matching and RPG for on-policy RL. It leverages differentiable ODEs for action generation, allowing end-to-end gradients through ODE solvers and dynamics without needing action log-likelihoods. The method features CFM regularization (for stability and exploration) and scales to action chunking architectures.

main contribution:
1. First demonstration that flow policies naturally fit the RPG framework, enabling direct backpropagation through ODE integration and dynamics without any log-likelihood computation.
2. On-policy RPG method for flow policies, plus two tailored CFM regularizers (past-data for stability, uniform-target for exploration) and an action-chunking extension.
3. Extensive results showing superior performance and sample efficiency compared to strong RPG and generative-policy baselines.

**Compliance With Llm Reviewing Policy:**

Affirmed.

**Final Justification:**

The author's response resolved my confusion.

**Key Questions For Authors:**

* How is the $D_{\text{recent}}$ buffer constructed?
* What would the ablation results look like under identical environmental conditions? Currently, the flow steps are insufficient, as we only have data for 5 and 6 steps.
* How does SHAC reduce the BPTT load? It seems to me that the real bottleneck lies in the flow steps rather than the horizon length.

**Limitations:**

yes

**Strengths And Weaknesses:**

Soundness:
The work offers a mathematically rigorous approach to flow policies through differentiable ODE integration, backed by an exceptionally thorough experimental suite across 7 diverse tasks.

Presentation:
The paper is extremely clear, well-structured.

Significance/Originality:
This work is interesting in the way it integrates flow matching with reparameterization gradients. However, the individual components are already well-established, and it’s unclear to me how RFO meaningfully differs from stochastic policies. While the key insight claims that “This structure is naturally compatible with the Reparameterization Policy Gradient framework,” I didn’t find any particularly compelling or standout advantages demonstrated.

---

> ### Author Rebuttal · Authors · 2026-03-30
>
> Thank you for your time and effort in reviewing our paper! We are very grateful for your constructive comments.
>
> ---
>
> #### **Significance & Originality**
>
> We respectfully clarify the contributions and advantages of RFO from the following perspectives:
>
> 1.  **From the RPG Perspective**: Introducing flow policies significantly enhances expressiveness, enabling the discovery of novel behaviors that Gaussian policies cannot capture. A concrete example is the **SoftJumper** task: RFO discovers **specialized high-jumping behaviors** (COM height reaching $5.7\times$ body height), whereas SAPO (Gaussian) does not find this mode. A snapshot of this behavior is available at this link: [JumpingBehavior](https://anonymous.4open.science/r/greatday/JumpingBehavior.png).
>
> 2.  **From the Flow RL Perspective**: RFO uniquely combines the strengths of distinct paradigms. It inherits the high sample efficiency of RPG while **bypassing intractable action log-likelihood computations**. Furthermore, as an on-policy algorithm, RFO requires significantly less training time than off-policy methods like **DrAC** and **FlowRL**.
>
> 3.  **Strong Empirical Performance**: By combining efficiency and expressiveness, RFO achieves SOTA results. Our aggregate training curve (IQM) further illustrates this at this link: [Aggregate Curve](https://anonymous.4open.science/r/greatday/aggregate_iqm_training_curve_preview.png).
>
> #### **Q1: Construction of $D_{recent}$**
>
> The $D_{recent}$ buffer stores on-policy rollout data from the **two most recent training iterations (including current iteration)**. At each iteration, the agent collects (observation, action) pairs through differentiable simulation, and the buffer retains only the current and previous iteration's data.
>
> Both are used as targets for the **Conditional Flow Matching (CFM)** loss:
> * **Current iteration actions** serve as targets to ensure the flow can reproduce its own behavior.
> * **Previous iteration actions** serve as a regularizer for the policy update, preventing drastic, unstable changes between iterations.
>
> #### **Q2: Ablations on Flow Steps**
>
> Following your suggestion, we conducted new experiments for RFO with flow steps **7, 8, 9, and 10** on the **SoftJumper** task. RFO maintains robust performance across a wide range of flow steps:
>
> | RFO's Flow Steps | Score (Mean) |
> | :--- | :--- |
> | 4 (Default) | 3023.96 |
> | 5 | 3088.27 |
> | 6 | 3351.77 |
> | 7 | 2873.81 |
> | 8 | 2982.68 |
> | 9 | 2929.92 |
> | 10 | 2712.93 |
>
> #### **Q3: Impact of Horizon (SHAC) vs. Flow Steps**
>
> SHAC is critical for RPG-based methods to avoid **gradient explosion** caused by backpropagating through long time horizons. To illustrate this, we extended RFO's horizon from **32** (default) to **128** on the **Ant** task.
>
> Performance **degrades from 5677 to 3660**, clearly demonstrating that a truncated horizon is essential for stability.
>
> In contrast, as shown in the ablation table above, RFO is highly robust to the number of flow steps.
>
> ---
>
> We hope these clarifications address your concerns. If so, we wonder if you could kindly consider **raising your score**? We will also be happy to answer any further questions you may have. Thank you very much!

---

> > ### Author Rebuttal · Reviewer_Eu4g · 2026-04-03
> >
> > I thank the authors for their detailed responses. While I have seen multiple works using flow as an actor, which makes the novelty feel somewhat limited, I acknowledge that I am not a specialist in this subfield. Based on the rebuttal and the feedback from other reviewers, I am considering increasing my score, though I will adjust my confidence accordingly.
> >
> > However, one minor question persists: why can SHAC effectively avoid the gradient explosion typically associated with backpropagation? In my view, the issue of exploding gradients is fundamentally rooted in the number of denoising teps. I remain curious about the authors' perspective on this.

---

> > > ### Author Response · Authors · 2026-04-03
> > >
> > > Thanks for the follow-up question. We would like to clarify the sources of gradient instability in BPTT.
> > >
> > >
> > > **Dynamics Jacobian.**
> > > The policy gradient requires backpropagation through the environment dynamics over the horizon $H$. The gradient $\partial s_T / \partial s_0$ involves a chain product of dynamics Jacobians:
> > >
> > > $$\frac{\partial s_T}{\partial s_0} = \prod_{t=0}^{T-1} \frac{\partial s_{t+1}}{\partial s_t} = \prod_{t=0}^{T-1} \left( \frac{\partial f}{\partial s_t} + \frac{\partial f}{\partial a_t} \frac{\partial a_t}{\partial s_t} \right)$$
> > >
> > > Physical dynamics often have spectral radius $\rho(\partial f / \partial s) > 1$ (due to contacts, collisions). This is precisely why SHAC truncates the horizon to $H = 32$, and why extending $H$ from 32 to 128 degrades performance.
> > >
> > > **Flow Jacobian.**
> > > For the flow policy, the action is generated via $K$-step Euler integration: $x_{k+1} = x_k + \frac{1}{K} v_\theta(x_k, t_k, s)$, and $a = x_K$. The gradient of the action w.r.t. the initial noise $x_0$ involves a chain product of flow Jacobians:
> > >
> > > $$\frac{\partial x_K}{\partial x_0} = \prod_{k=0}^{K-1} \frac{\partial x_{k+1}}{\partial x_k} = \prod_{k=0}^{K-1} \left( I + \frac{1}{K} \frac{\partial v_\theta}{\partial x_k} \right)$$
> > >
> > > In contrast to the dynamics chain, the flow chain has its own structure: each per-step Jacobian $I + \frac{1}{K} \frac{\partial v_\theta}{\partial x}$ involves an identity matrix, so its spectral radius is unlikely to be extremely greater than 1. Moreover, $K$ is small (4 in our default setting), and increasing $K$ correspondingly shrinks the Euler step size $\frac{1}{K}$.
> > >
> > > **Therefore, the horizon $H$—which controls the length of the dynamics chain—is the critical factor for gradient stability, not the number of flow steps $K$.** This is also clear from the computational graph: the flow Jacobians are nested inside each dynamics step and accumulate over different environment steps as well.
> > >
> > > Our ablation experiments confirm this analysis: varying flow steps from 4 to 10, the performance remains stable; while increasing the horizon from 32 to 128, a clear degradation is observed.

---

### Official Review · Reviewer_GaP4 · 2026-03-13

**Soundness:** 3
**Presentation:** 2
**Significance:** 3
**Originality:** 3
**Overall Recommendation:** 5
**Confidence:** 4

**Summary:**

This paper proposes Reparameterization Flow Policy Optimization (RFO), which introduces Reparameterization Policy Gradient to flow-based RL policies. In optimization, it backpropagates jointly through the flow generation process and environment dynamics. The paper further proposes two regularizers for training stability and exploration. Action chunking is also applied to RFO. Experiments on several locomotion and manipulation benchmarks demonstrate the effectiveness of RFO.

**Compliance With Llm Reviewing Policy:**

Affirmed.

**Final Justification:**

Thanks for the authors' reply. From the experiments, its efficiency is competitive. My main concern has been addressed. Thus I think this paper could be accepted. I will raise the score from 4 to 5.

**Key Questions For Authors:**

My main concerns are listed in the above weaknesses. The authors need to supplement the memory and time overhead in both the training and inference phases.

The presentation should be improved with consistent notations.

Some potential improvement: The assumptions about deterministic environment dynamics in the second paragraph of Section 3.3 are better to be moved to Section 3.1. They are all about the background and assumptions of RL.

**Limitations:**

Discussion about limitation is not presented. I think the main limitation could be the training and inference cost.

**Strengths And Weaknesses:**

**Strengths**
1. Overall, the idea of paper is novel. It combines flow-based RL policies with Reparameterization Policy Gradient. Compared with the traditional Reparameterization Policy Gradient, it breaks through the limitations of the Gaussian policies and obtains a more expressive action space.

2. The experimental results are good. It presents strong generalization ability on several tasks, outperforming the comparison RPG and Flow/Diffusion RL Baselines.

3. The writing is generally clear and easy to understand.

**Weaknesses**

1. My main concern is that the method seems to have high computational cost, as it requires backpropagation through both the flow generation process and environment dynamics, which may lead to large memory and time overhead. The authors only briefly analyze the efficiency of DrAC and RFO in Section 5.2, but there is a lack of data support. The authors need to supplement the memory and time overhead in both the training and inference phases.

2. In terms of writing, there are many detailed errors in notation. According to the definition of $\phi$ in Line 153, the order of $u$ and $x$ in the $v_\theta$ function is incorrect in several places in the subsequent text, including Line 155, Eq (4), (5), (8), (12), and (13). Also, in Line 160, $\phi(0,x_0)$ should be $p_0(x_0)$.

---

> ### Author Rebuttal · Authors · 2026-03-30
>
> Thank you for your time and effort in reviewing our paper! We are very grateful for your constructive comments.
>
> ---
>
> #### **W1, Concern 1 & Q1: Memory and Wall-clock Time**
>
> Thanks for this great question! To rigorously and fairly investigate the memory overhead, computational time, and inference time, we conducted new experiments on machines with **4 A40 GPUs**. The entire training and simulations were performed on GPUs. We ran 4 seeds each for **RFO (Ours)**, **DrAC**, **FlowRL**, and **SAPO** (SOTA RPG method) on the **SoftJumper** task (visual input), with each seed using one GPU.
>
> The results, including GPU memory usage, wall-clock training time, policy inference time, and average environment steps per second, are summarized below (reported as **mean [min, max]**):
>
> | Method | Inference Time (ms) | Training Throughput (steps/sec) | Training Time (h) | GPU Memory (GB) |
> | :--- | :--- | :--- | :--- | :--- |
> | **SAPO** | 0.97 [0.94, 1.21] | 727 [710, 743] | 2.36 [2.31, 2.40] | 4.23 |
> | **FlowRL** | 1.22 [1.19, 1.40] | 56 [56, 57] | 30.41 [30.26, 30.60] | 1.50 |
> | **RFO (Ours)** | **2.83 [2.74, 2.96]** | **635 [629, 642]** | **2.68 [2.66, 2.70]** | **7.10** |
> | **DrAC** | 8.73 [8.39, 9.50] | 107 [105, 108] | 16.04 [15.89, 16.20] | 2.33 |
>
> *Values in brackets denote [min, max] across seeds.*
>
> **Efficiency Analysis:**
> * **Training Time**: Among the three generative-model-based methods (RFO, FlowRL, and DrAC), RFO is by far the most efficient in training: it is **11x faster** than FlowRL (2.68h vs. 30.41h) and **6x faster** than DrAC (2.68h vs. 16.04h). RFO's training time (2.68h) is comparable to the Gaussian baseline SAPO (2.36h), with only **0.32h** additional overhead.
> * **Memory Trade-off**: RFO uses 7.10 GB of GPU memory. While higher than other methods, this is only **14.8%** of an A40's 48GB capacity and well within the range of consumer-grade GPUs (e.g., RTX 3080 with 10GB). We consider this a reasonable trade-off for the training time, memory efficiency, and algorithmic performance.
> * **Inference Latency**: RFO requires 2.83 ms per action (4-step flow integration), which is **3.1x faster** than DrAC (8.73 ms, 20-step DDPM denoising).
>
> #### **W2 & Q2: Notations and Writing**
>
> Thanks for helping us find these notation errors. We are very grateful for this and will correct them (including the order of $u$ and $\theta$, and the definition of $p_0(x_0)$) in the revised version.
>
> #### **Q3: Potential Improvement**
>
> Thanks for this constructive comment! Moving the deterministic environment assumptions to Section 3.1 is indeed a good suggestion, and we will follow it in the revised paper.
>
> ---
>
> We hope these clarifications address your concerns. If so, we wonder if you could kindly consider **raising your score**? We will also be happy to answer any further questions you may have. Thank you very much!

---

> > ### Author Rebuttal · Reviewer_GaP4 · 2026-04-03
> >
> > Thanks for the authors' reply. From the experiments, its efficiency is competitive. Thus I think this paper could be accepted. I will raise the score from 4 to 5.

---

> > > ### Author Response · Authors · 2026-04-03
> > >
> > > We are very happy to receive your recognition and would like to share our gratefulness for your time and effort in reviewing our work, including helping us fix the notations and writing. Thanks again!

---

### Decision · Program_Chairs · 2026-04-30

**Decision:**

Accept (regular)

**Comment:**

The reviewers and I are in agreement that the paper presents a reasonable combination of flow-based RL and reparameterization policy gradient. Methodologically, backpropping and unrolling through the flow steps is straightforward but can be memory and compute intensive. But, the paper finds a setting where it still works and shows modest but convincing results in comparison to all of the tasks considered in SHAC and related methods.